# Safety assessment of multiple systemic administration of human mesenchymal stem cell-conditioned medium for various chronic diseases

**Norihito Inami** *

Seihoku Clinic, 775 Takawa, Oshibedani, Nishi-ku, Kobe, Hyogo, Japan

* inami@shhc.jp

## Abstract

Conditioned medium (CM) derived from human mesenchymal stem cells (MSCs) has shown potential as a therapeutic agent. However, the safety of its administration in human remains largely unexplored. This study evaluated the safety of multiple systemic administrations of MSC-CM, specifically adipose-derived and umbilical cord-derived MSC-CM, in 55 patients with various chronic diseases. Symptom assessments and blood tests were conducted before and after administration to monitor adverse events and measure the inflammatory marker C-reactive protein (CRP), respectively. The results demonstrated no serious adverse events attributed to MSC-CM administration. Although minor adverse events were observed, their causal relationship with MSC-CM remained unclear. Additionally, MSC-CM administration slightly reduced CRP levels, regardless of the administration route (intraarterial, intravenous, or inhalation). Additionally, a significant reduction in CRP levels was observed in patients with elevated CRP levels (CRP > 0.3) following MSC-CM administration. These findings suggest that repeated systemic administration of MSC-CM is likely safe and may have anti-inflammatory effects.

## Introduction

Inflammation is a crucial protective response to pathogens that cause infection and tissue damage [1]. However, it is also implicated in various pathophysiological conditions, including heart failure [2], osteoarthritis [3], rheumatoid arthritis [4], inflammatory bowel disease [5], diabetes [6], acute lung injury [7], acute kidney injury [8], and tissue fibrosis [9]. Moreover, inflammation appears to contribute to fatigue and is closely associated with pain [10]. Given its central role in these diseases, anti-inflammatory therapies have been explored as potential treatments for multiple conditions.

Mesenchymal stem cells (MSCs), one of the most abundant types of somatic stem cells, have gained attention for their immunomodulatory and anti-inflammatory properties. MSCs regulate immune responses through various mechanisms, including the

**Data availability statement:** The data cannot be shared publicly because it contains personal information of patients. Data are available from the Seihoku Clinic Ethics Committee for researchers who meet the criteria for access to confidential data, through the contact person, Dr. Satoru Kimura, Chair of the Seihoku Clinic Ethics Committee (Clinical Engineering Technician, Seihoku Clinic) at kimura@shhc.jp. The contact person is responsible for storing and managing the data obtained from patients but is not involved in the content of the data or the writing of the manuscript.

**Funding:** The author(s) received no specific funding for this work.

**Competing interests:** The author has declared that no competing interests exist.

secretion of anti-inflammatory cytokines (e.g., IL-10, TGF-β, IL-1RA) and modulation of immune cell activity. Due to these properties, MSCs have been investigated in both preclinical and clinical settings for the treatment of inflammation-related diseases [11]. Among MSCs, adipose-derived MSCs (AD-MSCs) and umbilical cord-derived MSCs (UC-MSCs) have demonstrated distinct therapeutic potentials depending on the target disease. AD-MSCs have been shown to be effective in cardiac and pulmonary diseases by reducing myocardial inflammation and promoting tissue repair through the secretion of anti-inflammatory cytokines [2]. They also regulate immune responses by modulating macrophage and T-cell activity, helping restore balance in inflammatory conditions such as acute respiratory distress syndrome (ARDS) [7,12]. Meanwhile, UC-MSCs have exhibited superior regenerative effects in kidney diseases, with studies indicating that their paracrine effects promote renal tissue repair [13] Comparative studies suggest that UC-MSCs possess stronger immunomodulatory and anti-inflammatory properties than AD-MSCs, making them particularly promising for kidney regeneration [14,15].

Given that many of the therapeutic effects of MSCs are mediated by their secreted factors rather than the cells themselves, an MSC-conditioned medium (MSC-CM) has emerged as a promising alternative approach [16]. Compared to direct MSC transplantation, MSC-CM therapy offers several advantages. One key benefit is the reduced risk of immune rejection. Stem cell transplantation carries the possibility of immune recognition and rejection, which can trigger inflammatory responses. In contrast, MSC-CM does not contain living cells, minimizing the likelihood of immune rejection and significantly lowering the risk of adverse immune reactions [17]. Another advantage is the reduced potential for tumorigenesis. While both endogenous and exogenous MSCs have been reported to support tumor cell proliferation in certain contexts, studies suggest that MSC-CM exerts tumor-suppressive effects by inhibiting oncogenic signaling rather than promoting it, underscoring its potential safety for patients at risk of cancer [18,19].

Nevertheless, as MSC-CM includes factors released from MSCs as well as those contained in the culture medium, these factors may comprehensively affect the safety and efficacy of MSC-CM. MSC cultures were conventionally conducted in a medium containing fetal bovine serum (FBS). However, culturing in a medium containing FBS resulted in MSC-CM containing animal-derived components, and the possibility of viral infection could not be ruled out. Additionally, there are several human clinical reports on the use of MSC-CM for various conditions [20–22], but the safety and efficacy of MSC-CM are thought to vary depending on the raw materials and manufacturing methods. Generalizing these results and using them for human treatment is dangerous. Furthermore, all these reports involved small local administration, such as subcutaneous, intramuscular, and intrathecal administration, and to the best of our knowledge, there are no reports regarding the safety of systemic administration.

To address these concerns, we aimed to reduce the risk of infection for patients receiving the treatment by collecting adipose tissue or umbilical cords from healthy donors with no infectious diseases and culturing them in an animal origin-free (AOF) medium, after which we selected an MSC-CM that had undergone thorough quality

control, such as virus-proof testing of the completed MSC-CM. Additionally, we summarized the safety of administering relatively large amounts of MSC-CM via multiple administration routes in four observational studies.

## Materials and methods

### Ethical considerations

This study was conducted in compliance with the Ethical Guidelines for Medical and Health Research Involving Human Subjects, and it was approved by our institution's ethical review board (Seihoku Clinic ethical review board accreditation number 22000152), after which it was registered in the Japan Registry of Clinical Trials (jRCT) (Table 1). The attending physician explained the research content to the research participants using an informed consent document and obtained their voluntary written consent to receive treatment.

### Participants

The participants were patients enrolled in an observational study at the Seihoku Clinic between October 1, 2022, and August 31, 2023. Participants for each target disease included individuals who did not achieve satisfactory treatment effects with standard therapies, those who opted against standard medications due to concerns about side effects or other adverse factors, and those for whom physicians deemed MSC-CM treatment to be a suitable option.

Patients aged 18 years or older with capacity to provide informed consent, who were able to provide sufficient explanation and voluntary written consent after receiving a consent explanation document regarding this treatment, and whom physicians deemed in need for treatment were included in the study. Patient who satisfied with any of the following conditions were excluded: those who had a history or suspicion of dementia, used recreational drugs or stimulants, or were pregnant or breastfeeding and those judged by their attending physicians to be unsuitable for this study.

Additional selection criteria included individuals diagnosed with lung disease alone, such as diffuse panbronchiolitis, interstitial lung disease, or chronic obstructive pulmonary disease (COPD); those experiencing difficulty breathing with a lung disease deemed suitable for treatment by a physician; individuals with heart disease alone whose serum N-terminal pro–B-type natriuretic peptide (NT-proBNP) levels were ≥ 900 pg/mL; or those with chronic renal failure alone whose estimated glomerular filtration rate (eGFR) was ≤ 40 mL/min/1.73 m².

### MSC-CM preparation

Human adipose-derived MSC-CM (AD-MSC-CM) and human umbilical cord-derived MSC-CM (UC-MSC-CM) were manufactured using an AOF medium; the manufacturing process was outsourced to BioMimetics Sympathies, Inc. (Tokyo, Japan). Tissue samples for AD-MSC-CM and UC-MSC-CM were obtained respectively from two Japanese women in their 20s who passed a virus-negative test and who provided a consent form.

**Table 1. List of observational study titles.**

| Study Title | Participant Code | Ethics Committee Approval Number | jRCT Registration Number |
|---|---|---|---|
| A study on the treatment of heart failure with human-derived mesenchymal stem cell conditioned medium | SH | 20220819-1 | jRCT1051220088 |
| A study on the treatment of general fatigue with human-derived mesenchymal stem cell conditioned medium | SM | 20220819-2 | jRCT1051220089 |
| A study on the treatment of lung diseases using human-derived mesenchymal stem cell conditioned medium | SL | 20220819-3 | jRCT1051220090 |
| A study on the treatment of chronic renal failure using human-derived mesenchymal stem cell-conditioned medium | SK | 20220819-4 | jRCT1051220091 |

AD-MSCs and UC-MSCs were isolated from the samples, and the stromal vascular fraction was passaged in MS-E0001 medium (BioMimetics Sympathies, Inc., Tokyo, Japan), a serum-free culture medium, at 37 °C in an atmosphere containing 5% $CO_2$. Then, 1.2-1.8 × $10^6$ cells of passage 4–5 AD-MSCs and UC-MSCs were seeded into each Corning® T175 CellBIND™ Culture Flask (Corning Inc., NY, USA) containing 28–35 mL of MS-E0001 medium (BioMimetics Sympathies, Inc.). When cell confluency reached 90%, the medium was changed to MS-E0006 medium (BioMimetics Sympathies, Inc.), and the cells were incubated for an additional 2 days. The resulting MSC-CM was collected and filtered through a 0.22 μm filter. A virus-free test was conducted during the manufacturing process of the conditioned medium, and the final products were subjected to endotoxin, sterility, mycoplasma, and color tests, all of which were cleared. We also conducted a visual inspection for foreign material contamination and confirmed the absence of such contaminations. MSC-CM was aliquoted and stored at -30 °C until use. AD-MSC-CM and UC-MSC-CM used in the present study were all MSCs that were respectively established from the same donor.

### Enzyme-linked immunosorbent assay

The measurement of the concentration of hepatocyte growth factor (HGF) in the CM was conducted using a Quantikine enzyme-linked immunosorbent assay (ELISA) kit (Human HGF Immunoassay, R&D Systems, Minneapolis, MN, USA) according to the manufacturer's protocol. The exosome amount in the CM was quantified using a "CD63/CD63 ELISA kit for human-derived exosome quantification" (HAK-HEL6363–1, Hakarel, Japan). The assay was conducted according to the manufacturer's protocol. HGF was measured as it is known to promote cell growth, migration, and differentiation and is an important factor for evaluating the therapeutic effects of regenerative medicine. Meanwhile, exosomes were investigated as they are involved in intercellular signaling and functional regulation and contribute to cell repair.

### Human umbilical vein endothelial cell assay

TNF-α induces inflammation and premature senescence in endothelial cells, such as human umbilical vein endothelial cells (HUVECs) [23]. HUVECs were purchased from the Japan Cell Research Bank (JCRB) and cultured in DMEM/F12 (supplemented with 10% FBS, 10 ng/mL b-FGF, and 10 μg/mL gentamicin) at 37 °C and 5% $CO_2$. As described elsewhere [23], once the cell density reached an appropriate level, 10 ng/mL TNF-α was added to induce inflammation. Simultaneously, the control MS-E0006 medium, AD-MSC-CM, or UC-MSC-CM was added to the medium to a final concentration of 75% and incubated for three days. The cells were then lysed, and RNA was extracted using ReliaPrep™ (Promega, Madison, WI, USA) according to the manufacturer's protocol.

DNA was prepared by reverse transcription (Toyobo Co., Ltd., Osaka, Japan). Real-time PCR was conducted using CFX Connect™ (Bio-Rad, Hercules, CA, USA) to evaluate the IL-6 and IL-8 expression levels as inflammation markers. GAPDH was used as an internal standard for correction.

### Administration schedule

Fig 1 shows the schedule for administration, blood tests, and related procedures. Patient interviews were conducted approximately 1 week before the first administration. For intra-arterial (IA) and intravenous (IV) administration, the CM was administered three times per month. For inhalation (INH) administration, the CM was administered once per week for a total of 12 sessions.

Lower panel: Participants' schedule for inhalation (INH) administration. Patients were interviewed approximately one week before the first administration, and 12 administrations were conducted weekly. Blood samples were collected immediately before the first, fifth, and ninth administrations, and one week after the 12th administration, with serum stored. Interviews were conducted before the third and seventh administrations, and one week after the 12th administration.

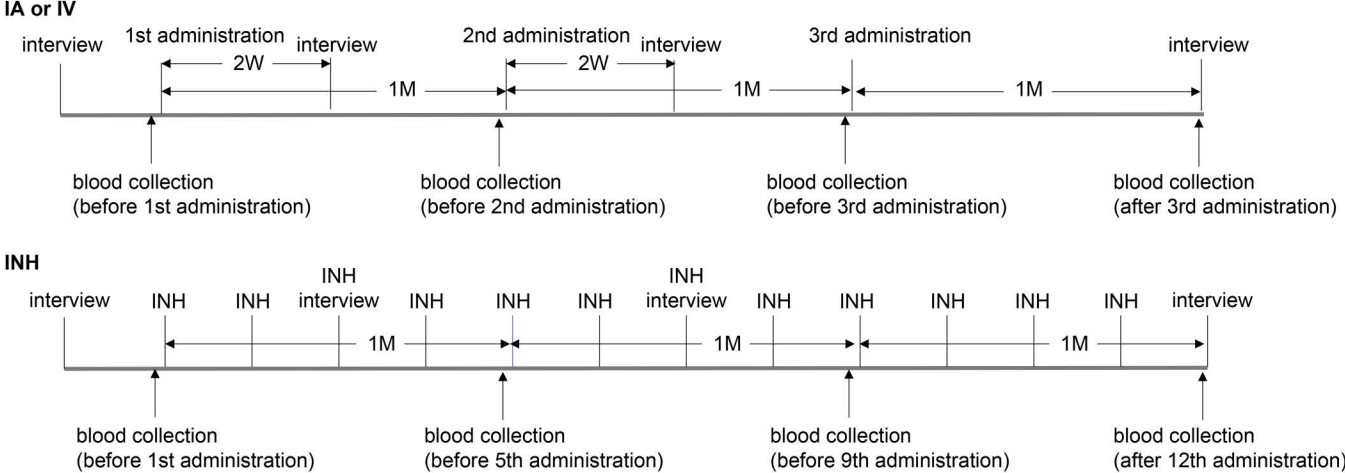

**Fig 1. Administration schedule.** Upper panel: Participants' schedule for intra-arterial (IA) or intravenous (IV) administration. Patients were interviewed approximately one week prior to the first administration, and the CM was administered three times per month. Blood samples were collected immediately before each administration and one month after the third administration, with serum stored. Interviews about changes in subjective symptoms were conducted two weeks after the first and second administrations, and one month after the third administration.

## Administration methods

**(1) Basic policy.** Cases where the target organ is clear, such as heart failure, lung disease, or kidney failure, involved a large amount of the CM being administered over long periods of time, with the expectation that a significant effect would be achieved by delivering it directly to the diseased site. Considering the burden that would be imposed on patients, we set the administration time to 24 hours. In a previous study, Dahbour, et al. involved administering an average of 18 mL of CM into the thecal cavity [21]. Considering that IA administration would result in more diffusion than thecal administration, we administered approximately double the amount (30 mL).

As the organ responsible for the disease is unknown in fatigue treatment, IV administration (IV) was conducted to ensure delivery throughout the body. Considering the economic burden on patients, we reduced the amount to 10 mL compared to IA administration. We also observed the patient's condition for rapid diffusion and safety by avoiding administration over a short period of time and conducting the administration over a three-hour period.

**(2) Lung disease and heart failure.** For patients with lung diseases or heart failure, we mixed 30 mL of AD-MSC-CM with 500 mL of physiological saline (Otsuka Pharmaceutical Co., Ltd., Tokyo, Japan). The mixture was injected into the pulmonary artery, which was accessed from the medial cubital vein using a Swan-Ganz Catheter (Model TD4L-1035C; Edwards Lifesciences Corp., Irvine, California, USA) and administered for 24 h. UC-MSC-CM was used for the next administration if no improvement in subjective symptoms was observed. An IV administration for 3 h was performed when catheter administration was difficult (Participant Codes: SL005, SH002, and SH005).

**(3) Chronic kidney disease.** For patients with chronic renal failure, we mixed 30 mL of UC-MSC-CM with 500 mL of physiological saline. The mixture was administered via a Good Tech HT Catheter (Good Tech Co., Ltd., Tokyo, Japan) inserted into the median cubital vein and administered throughout approximately 24 h into the descending aorta, 2 cm proximal to the renal artery bifurcation.

**(4) General fatigue.** For patients with general fatigue, we mixed 10 mL of AD-MSC-CM with 500 mL of physiological saline. The mixture was infused into the median cubital vein over approximately 3 hours. UC-MSC-CM was administered at the next administration if no improvement in subjective symptoms was observed. Inhalation administration (INH) was administered if IV administration was difficult to perform (Participant Code: SM019). An INH was performed with a mix of

1 mL of AD-MSC-CM and 4 mL of physiological saline using a nebulizer (Shenzhen IMDK Medical Technology Co. Ltd., China) once per week for a total of 12 times.

### Interviews

Before administration, the attending physician inquired about the patients' symptoms and past medical history. The attending physician conducted interviews regarding the patients' subjective symptoms, including adverse events, 2 weeks after the first and second administrations and 1 month after the third administration for IA or IV. For INH, interviews were conducted on the days of the third and seventh administrations and 1 week after the 12th (final) administration (Fig 1).

### Blood tests

Serum was collected before each administration and 1 month after the third administration (immediately before the first, fifth, and ninth administrations, and 1 week after the 12th administration for INH), with the necessary tests conducted for each disease.

While MSCs have immunomodulatory effects [24,25], MSC-CM administration may provoke inflammation due to unexpected allergic reactions or other adverse responses. Therefore, it is crucial to monitor the inflammatory levels in patients during the investigation. To assess inflammation, we measured serum C-reactive protein (CRP) levels before and after MSC-CM administration. CRP is primarily produced in the liver and is rapidly secreted in response to inflammation in the body [26]. Blood test items were specified in each research protocol depending on the target disease for which the study participant was entered. Among them, CRP levels are a common test item for the four diseases; therefore, we evaluated CRP levels in all participants. CRP-level evaluations were outsourced to BML Inc. (Tokyo, Japan).

### Statistical analyses

Kolmogorov–Smirnov test was performed to assess the normality of serum CRP data before and after administration of the CM. As the data did not follow Gaussian distribution, a two-tailed Wilcoxon matched-pairs signed rank test was performed. Statistical analyses were performed using R version 4.4.1 (R Foundation for Statistical Computing, Vienna, Austria). A significance level of 0.05 was chosen as the p-value cutoff, with p-values less than 0.05 considered statistically significant.

## Results

### Analyses of the features of AD- and UC-MSC-CM

To compare the features of AD-MSC-CM and UC-MSC-CM, we first analyzed the amounts of HGF and exosomes using ELISA. The results showed no significant difference in the amounts of HGF and exosomes between AD-MSC-CM and UC-MSC-CM (Fig 2).

Next, we investigated the anti-inflammatory effect of MSC-CM using HUVECs (Fig 3). The results showed that treatment with both AD-MSC-CM and UC-MSC-CM suppressed TNF-α-induced *IL-6* and *IL-8* mRNA expression levels, thereby suggesting that both AD-MSC-CM and UC-MSC-CM exhibit anti-inflammatory activity.

### Safety tests

There were 55 study participants (31 men, 24 women, mean age 77.4 ± 10.1 years). The number of entries was as follows: lung disease, 12 (7 men, 5 women); heart failure, 14 (7 men, 7 women); chronic renal failure, 10 (7 men, 3 women); and general fatigue, 19 (10 men, 9 women).

There were 54 participants who received 3 intravascular administrations and 1 participant who received 12 INH, for a total of 174 administrations (Table 2). Minor adverse events were observed, including dizziness during a massage

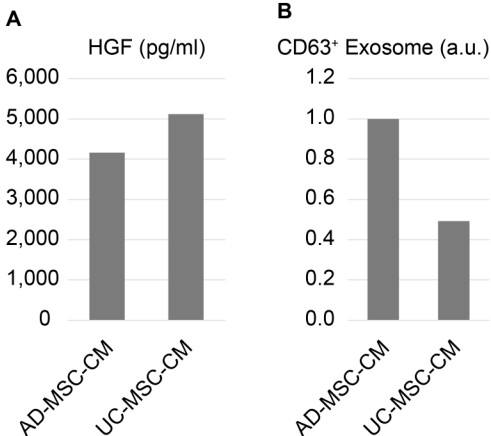

**Fig 2. HGF and Exosome Levels in MSC-CM. (A)** HGF concentration in AD-MSC-CM and UC-MSC-CM was measured. **(B)** Exosome quantity in AD-MSC-CM and UC-MSC-CM was assessed. Exosome levels are reported as relative values.

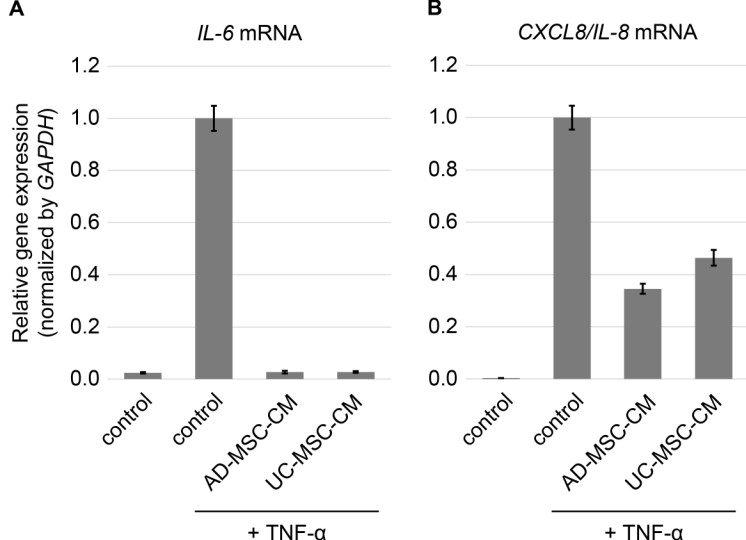

**Fig 3. Expression of Pro-Inflammatory Cytokine IL-6 and IL-8 in HUVECs.** Expression of *IL-6* mRNA (**A**) and *CXCL8/IL-8* mRNA (**B**) in inflammation-tion induced HUVECs after treatment with MSC-CM. HUVECs were treated with control medium (control), AD-MSC-CM, or UC-MSC-CM (75%) with or without TNF-α (10 ng/mL) for three days. Data are presented as mean ± SEM.

**Table 2. Administration Route and Types of MSC-CMs.**

|  | PA (30 mL) | DAo (30 mL) | IV (10 mL) | IV (30 mL) | INH (1 mL) |
|---|---|---|---|---|---|
| AD-MSC-CM | 63 | 0 | 47 | 9 | 12 |
| UC-MSC-CM | 6 | 30 | 7 | 0 | 0 |

PA, pulmonary artery administration, DAo; descending aorta administration, IV, intravenous administration; INH, inhalation administration; MSC-CM, mesenchymal stem cell-conditioned medium.

(SK003, after the first administration, Common Terminology Criteria for Adverse Events (CTCAE) version 5.0 code 1001375, Grade 1), anorexia (SK004, after the second administration, CTCAE code 10002646, Grade 1), enterocolitis (SH008, after the third administration, CTCAE code 10014893, Grade 1), post-renal hydronephrosis (SH008, after the third administration, CTCAE code 10038369, Grade 2), and a fever of 40 °C (SH009, after the first administration, CTCAE code 10016558, Grade 2). No serious adverse events, such as death, life-threatening conditions, significant disability or permanent damage, or the need for urgent medical intervention, were observed during any of the 174 administrations. Of the 55 total patients who received MSC-CM administration, 4 experienced minor adverse reactions of unknown causality related to MSC-CM administration.

## Changes in inflammatory markers

We found no statistically significant difference in serum CRP levels before and after the administration of both AD- and UC-MSC-CM (Fig 4A; p = 0.5994, two-tailed Wilcoxon matched-pairs signed-rank test, n = 55). CRP levels slightly decreased after administration, suggesting that MSC-CM administration may not provoke inflammatory responses.

Next, to evaluate the effect of the administration route, we compared the changes in CRP levels before and after MSC-CM administration for each route (Fig 4B-4E). Consistent with our previous result (Fig 4A), CRP levels tended to decrease across all administration routes, but the changes were not significant. Interestingly, INH administration appeared to reduce CRP levels more robustly, although there was only one such patient in this study.

Notably, patient SH008 developed unexplained enterocolitis (CRP: 14.1 mg/dL) 2 days after the third PA administration for an unknown reason, and was placed under special observation. Two days later, the patient was diagnosed with post-renal hydronephrosis (CRP: 8.0 mg/dL) and received treatment at the Urology Department of the Nishi-Kobe Medical Center. The post-administration blood test for patient SH008 in this study was conducted 36 days after

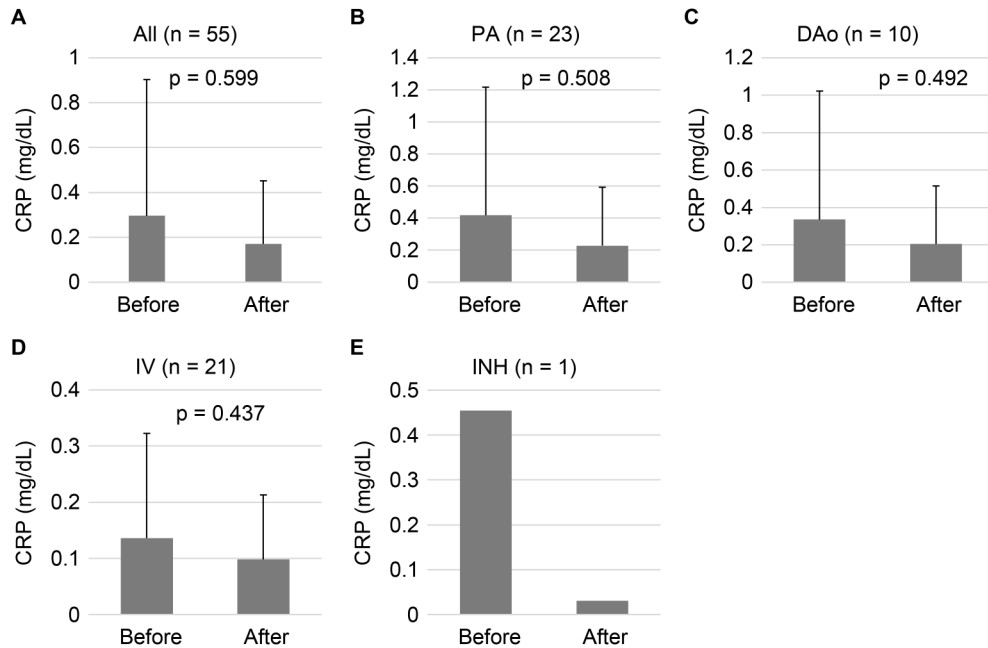

**Fig 4. Changes in CRP Levels Before and After Administration of MSC-CM.** CRP levels (mg/dL) in serum were measured before and after the administration of MSC-CM in all 55 patients **(A)**, 23 for PA **(B)**, 10 for DAo **(C)**, 21 for IV **(D)**, and 1 for INH **(E)**. The data are mean ± SD. Statistical analysis was performed using a two-tailed Wilcoxon matched-pairs signed rank test. PA, pulmonary artery administration, DAo; descending aorta administration, IV, intravenous administration; INH, inhalation administration.

the diagnosis of post-renal hydronephrosis. It remains unclear whether this adverse event was caused by MSC-CM administration.

Due to significant variation in the pre-administration CRP values of the CM (average 0.296 ± 0.607 mg/dL), we divided and analyzed the data based on the CRP values. According to Nehring et al.'s criteria [27], values less than 0.3 mg/dL are classified as "Normal" status (a level seen in most healthy adults). Therefore, we divided the CRP values into two groups: 0.3 or higher (CRP > 0.3) and below 0.3 (CRP < 0.3). We found that MSC-CM administration significantly decreased CRP levels in patients with CRP > 0.3 (Fig 5A; 1.109 to 0.413, p = 0.0122, two-tailed Wilcoxon matched-pairs signed-rank test, n = 12), suggesting that MSC-CM administration resulted in anti-inflammatory responses in patients with elevated inflammatory status. On the other hand, CRP levels significantly increased in patients with CRP < 0.3 (Fig 5B; 0.069 to 0.098, p = 0.0415, two-tailed Wilcoxon matched-pairs signed-rank test, n = 43). However, the post-administration CRP level (0.098 ± 0.113) remained below 0.3, indicating that MSC-CM may not elevate the inflammatory response in patients with "Normal" status.

## Discussion

The primary objective of this study was to confirm the safety of multiple systemic administrations of relatively large volumes (10 or 30 mL) of MSC-CM. No significant differences were observed in the contents or effects of AD-MSC-CM and UC-MSC-CM within the scope of our investigation. Therefore, it was not possible to determine which should be preferentially used for specific diseases.

We observed no serious adverse events in patients who had MSC-CMs intravascularly administered (3 times each to 54 patients totaling 162 administrations) or INH (12 times to 1 patient). Considering the immune system's reaction mechanisms, it is possible that the first administration could sensitize the immune system, with subsequent administrations potentially leading to an excessive immune response, such as anaphylactic shock. However, no excessive immune responses were observed after the second administration in this study. This may be because the components in MSC-CM either elicit an appropriate immune response or do not significantly stimulate the immune system, thus preventing an excessive reaction. Our findings suggest that both AD-MSC-CM and UC-MSC-CM can be safely administered multiple times.

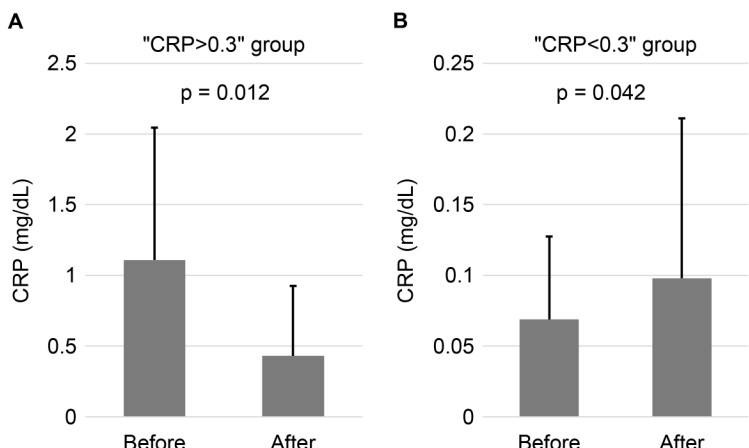

**Fig 5. Changes in CRP Levels Before and After Administration of MSC-CM. (A)** Group with pre-administration CRP levels below 0.3 mg/dL. **(B)** Group with pre-administration CRP levels above 0.3 mg/dL. CRP levels (mg/dL) in serum were measured before and after MSC-CM administration in 55 individuals. Data are presented as mean ± SD. Statistical analysis was performed using a two-tailed Wilcoxon matched-pairs signed-rank test.

Minor adverse events were reported in four patients. Two patients with chronic renal failure who received UC-MSC-CM via the descending aorta experienced minor adverse events such as anorexia and vertigo. Additionally, one patient with heart failure who received AD-MSC-CM via the pulmonary artery developed a transient fever. These symptoms occurred only once during the three administrations per patient. CRP levels in these patients either decreased (SK004) or fluctuated within 0.3 mg/dL (SK003 and SH009). Furthermore, all symptoms resolved within 1–2 days without medical intervention, suggesting that MSC-CM was unlikely the cause and that these events were incidental.

Another patient (SH008), who had a baseline CRP level above 0.3 mg/dL, experienced an increase in CRP after MSC-CM administration. Two days after the third administration of AD-MSC-CM via the pulmonary artery, the patient was diagnosed with unexplained enterocolitis, followed by post-renal hydronephrosis. However, given the administration route, these conditions were unlikely to be directly caused by MSC-CM. While these findings do not negate the overall safety of MSC-CM, further investigations are necessary to establish conclusive evidence.

This study evaluated the safety of systemic MSC-CM administration by monitoring changes in CRP levels. Overall, the average CRP level decreased across all 55 patients. Notably, in the group with CRP levels of 0.3 mg/dL or higher, MSC-CM administration led to a significant reduction in CRP. These findings suggest that MSC-CM may have the potential to suppress systemic inflammation. In patients with baseline CRP levels below 0.3 mg/dL, there was a statistically significant increase in CRP levels. However, even the highest CRP level in this group remained below 0.5 mg/dL. According to Nehring et al. [27], CRP levels between 0.3 and 1.0 mg/dL are considered normal or may reflect minor elevations due to factors such as obesity, pregnancy, depression, diabetes, common colds, gingivitis, periodontitis, a sedentary lifestyle, smoking, or genetic polymorphisms. Therefore, this increase was not considered clinically significant.

The average age of the participants was 76.2 ± 11.1 years, further supporting the safety of MSC-CM in older patients. The mild nature of the reported adverse events reinforces the safety of systemic MSC-CM administration. These findings suggest that MSC-CM is a promising treatment option for chronic diseases, offering physicians a potential new therapeutic strategy.

Few studies regarding the human administration of MSC-CM have been previously reported, with the only reports of MSC-CM-derived exosomes (trade name ExoFlo, Direct Biologics, Texas, USA) administered intravenously at doses of up to 15 mL [28,29]. To the best of our knowledge, local administration is the only method of MSC-CM administration that has not been commercialized [20–22,30–51]. Our report is extremely rare among the reports related to the safe administration of multiple systemic administrations of relatively large volumes (30 mL) of MSC-CM.

In the present study, MSC-CM was mainly administered through the pulmonary artery, descending aorta, or IV route. Administration via the artery is CT-guided and requires advanced equipment and techniques; however, in the present study, there was no clear difference in serum CRP level reduction in the IV or INH groups compared to IA administrations, suggesting that this technique could be widely and generally used.

This study has some limitation. First, INH was performed in only one patient receiving treatment for general fatigue. This patient was initially scheduled for IV administration but requested to avoid IV treatment, leading the attending physician to switch to INH. As a result, the number of INH cases in this study was limited. Although an improvement in CRP was observed in this case, the sample size was too small to compare INH with other administration routes. Future studies should include more cases to better evaluate the potential inhibitory effect of INH on CRP levels. Additionally, assessing efficacy using test parameters beyond CRP levels will be an important focus for future research. Furthermore, examining changes in antioxidant capacity, Galectin-3, and NT-proBNP levels in patients with heart diseases, as well as eGFRs in patients with renal diseases, could be beneficial. The sample size of this study was insufficient for comparing differences in effectiveness due to the different administration routes for each disease. Future research requires increasing the sample size and studying whether similar effects can be obtained with methods that place less of a burden on patients.

In conclusion, this study confirmed the safety of multiple MSC-CM administrations manufactured using AOF technology in 55 patients with chronic diseases unresponsive to conventional treatments. No severe adverse events were observed,

and minor adverse events were limited to a few cases. Changes in CRP levels indicated that systemic MSC-CM administration did not induce inflammatory responses. These findings suggest that properly manufactured and administered MSC-CM may be a promising treatment for chronic diseases, providing physicians and patients with a potential new therapeutic option. Further research is needed to fully explore its therapeutic potential.

## Acknowledgments

We would like to thank Editage (www.editage.jp) for the English language editing.

## Author contributions

**Conceptualization:** Norihito Inami.

**Data curation:** Norihito Inami.

**Formal analysis:** Norihito Inami.

**Investigation:** Norihito Inami.

**Methodology:** Norihito Inami.

**Project administration:** Norihito Inami.

**Resources:** Norihito Inami.

**Software:** Norihito Inami.

**Supervision:** Norihito Inami.

**Validation:** Norihito Inami.

**Visualization:** Norihito Inami.

**Writing – original draft:** Norihito Inami.

**Writing – review & editing:** Norihito Inami.

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
