## [Decision Letter · Decision Letter 0]

20 Aug 2024

Dear Dr. Inami,

Thank you for submitting your manuscript to PLOS ONE. After careful consideration, we feel that it has merit but does not fully meet PLOS ONE’s publication criteria as it currently stands. Therefore, we invite you to submit a revised version of the manuscript that addresses the points raised during the review process.

The mesenchymal stem cell-conditioned medium gives promising results in in vitro and in vivo studies, and it is exciting to see that these mediums can be applied to humans. Therefore, the presented work is valuable, but some requirements must be met in order to be publishable.

Introduction

1. The introduction is weak for this study. The importance of using the conditioned medium derived from mesenchymal stem cells for inflammatory diseases should be emphasized. The literature contains many in vitro and in vivo studies on this subject. The introduction should be supported by these studies. Why is it desired to be used in humans? This context should be well established.

Methods

2. How many donors were used as sources of mesenchymal stem cells? Was CM prepared separately from the cells obtained from these donors? Which source was the CM used for which disease? It should be specified.

3. Has the content of CM obtained from adipose tissue been compared with that of CM obtained from the umbilical cord? This difference is significant, and the study should have been planned with this in mind.

4. The preparation conditions of CMs should be explained in the method section, such as the number of cells, how long they were kept, how much medium, medium, and supplement content were used, etc.

5. How were the doses determined? Your sample size for INH is 1. This is insufficient.

Results

6. What do 1 and 2 represent in Figure 2? The x-axis naming of all graphs should be corrected. Label in an understandable way when looking at the graph or add a description to the figure legend. It is irrational for the "error bar SE" label to be on the x-axis. Add this to the figure legend and label it in a way that makes your data understandable.

Discussion

7. The findings should be supported by literature data in the discussion. Possible safety and side effects should be discussed with possible reasons, considering variables such as administration routes, CM source used, disease, etc.

Thanks

Shaifur Rahman, PhD

We look forward to receiving your revised manuscript.

Kind regards,

Md Shaifur Rahman, Ph.D

Academic Editor

PLOS ONE

Additional Editor Comments:

Dear Authors,

Please address the following points raised by the reviewer.

The mesenchymal stem cell-conditioned medium gives promising results in in vitro and in vivo studies, and it is exciting to see that these mediums can be applied to humans. Therefore, the presented work is valuable, but some requirements must be met in order to be publishable.

Introduction

1. The introduction is weak for this study. The importance of using the conditioned medium derived from mesenchymal stem cells for inflammatory diseases should be emphasized. The literature contains many in vitro and in vivo studies on this subject. The introduction should be supported by these studies. Why is it desired to be used in humans? This context should be well established.

Methods

2. How many donors were used as sources of mesenchymal stem cells? Was CM prepared separately from the cells obtained from these donors? Which source was the CM used for which disease? It should be specified.

3. Has the content of CM obtained from adipose tissue been compared with that of CM obtained from the umbilical cord? This difference is significant, and the study should have been planned with this in mind.

4. The preparation conditions of CMs should be explained in the method section, such as the number of cells, how long they were kept, how much medium, medium, and supplement content were used, etc.

5. How were the doses determined? Your sample size for INH is 1. This is insufficient.

Results

6. What do 1 and 2 represent in Figure 2? The x-axis naming of all graphs should be corrected. Label in an understandable way when looking at the graph or add a description to the figure legend. It is irrational for the "error bar SE" label to be on the x-axis. Add this to the figure legend and label it in a way that makes your data understandable.

Discussion

7. The findings should be supported by literature data in the discussion. Possible safety and side effects should be discussed with possible reasons, considering variables such as administration routes, CM source used, disease, etc.

Thanks

Shaifur Rahman, PhD

Reviewers' comments:

Reviewer's Responses to Questions

**Comments to the Author**

1. Is the manuscript technically sound, and do the data support the conclusions?

Reviewer #1: Yes

2. Has the statistical analysis been performed appropriately and rigorously?

Reviewer #1: Yes

3. Have the authors made all data underlying the findings in their manuscript fully available?

Reviewer #1: No

4. Is the manuscript presented in an intelligible fashion and written in standard English?

Reviewer #1: Yes

Reviewer #1: The mesenchymal stem cell-conditioned medium gives promising results in in vitro and in vivo studies, and it is exciting to see that these mediums can be applied to humans. Therefore, the presented work is valuable, but some requirements must be met in order to be publishable.

Introduction

1. The introduction is weak for this study. The importance of using the conditioned medium derived from mesenchymal stem cells for inflammatory diseases should be emphasized. The literature contains many in vitro and in vivo studies on this subject. The introduction should be supported by these studies. Why is it desired to be used in humans? This context should be well established.

Methods

2. How many donors were used as sources of mesenchymal stem cells? Was CM prepared separately from the cells obtained from these donors? Which source was the CM used for which disease? It should be specified.

3. Has the content of CM obtained from adipose tissue been compared with that of CM obtained from the umbilical cord? This difference is significant, and the study should have been planned with this in mind.

4. The preparation conditions of CMs should be explained in the method section, such as the number of cells, how long they were kept, how much medium, medium, and supplement content were used, etc.

5. How were the doses determined? Your sample size for INH is 1. This is insufficient.

Results

6. What do 1 and 2 represent in Figure 2? The x-axis naming of all graphs should be corrected. Label in an understandable way when looking at the graph or add a description to the figure legend. It is irrational for the "error bar SE" label to be on the x-axis. Add this to the figure legend and label it in a way that makes your data understandable.

Discussion

7. The findings should be supported by literature data in the discussion. Possible safety and side effects should be discussed with possible reasons, considering variables such as administration routes, CM source used, disease, etc.

**Do you want your identity to be public for this peer review?** For information about this choice, including consent withdrawal, please see our Privacy Policy

Reviewer #1: No

---

## [Author Response · Author response to Decision Letter 1]

1 Nov 2024

Response to Reviewers

1. The introduction is weak for this study. The importance of using the conditioned medium derived from mesenchymal stem cells for inflammatory diseases should be emphasized. The literature contains many in vitro and in vivo studies on this subject. The introduction should be supported by these studies.

We have introduced new reviews and articles for heart failure and lung disease to the introduction. Many in vitro reports are cited in these studies, and treatment with adipose-derived stem cells has been shown to be effective. Several articles were also cited for kidney failure. These articles have shown that umbilical cord-derived mesenchymal stem cells (MSCs) are effective, and the main effect is thought to be the anti-inflammatory effect of factors released from the cells.

Why is it desired to be used in humans? This context should be well established.

We described that stem cell therapy is widely used, compared stem cell therapy with treatment using the conditioned medium, showed that treatment using the conditioned medium has many advantages from both the patient and manufacturing perspectives, and provided the justification for starting conditioned medium treatment in patients.

Methods

2. How many donors were used as sources of mesenchymal stem cells? Was CM prepared separately from the cells obtained from these donors?

We have included additional descriptions of the donors and CM preparation under the headings “Conditioned medium used for treatment,” and “Comparison on MSC-CMs.

Which source was the CM used for which disease? It should be specified.

The conditioned medium that was used for each disease was described in the methods under the heading, “dosing methods.”

3. Has the content of CM obtained from adipose tissue been compared with that of CM obtained from the umbilical cord? This difference is significant, and the study should have been planned with this in mind.

We included in vitro experimental data to strengthen the justification for the type of conditioned medium selected.

4. The preparation conditions of CMs should be explained in the method section, such as the number of cells, how long they were kept, how much medium, medium, and supplement content were used, etc.

Thank you for your comment. We have included additional descriptions in the methods to descript the preparation conditions.

5. How were the doses determined? Your sample size for INH is 1. This is insufficient.

Results

For heart failure and lung disease cases, administration was conducted into the pulmonary artery via catheter. For kidney failure, administration was conducted into the descending aorta near the renal artery. Given the direct administration to the diseased site, we administered a large amount of the conditioned medium over a long period of time with the hope of seeing a significant effect. Considering the burden imposed on patients, we set an administration time of 24 hours. In Dahbour et al., (2017), , they administered an average of 18 mL of conditioned medium into the spinal cavity. Considering that arterial administration has more diffusion than thecal administration, we administered the approximately doubled amount (30 mL).

The organ causing the disease is unknown for fatigue and chronic pain treatment, so intravenous administration was conducted with the aim of systemic diffusion. Considering the economic burden on patients, the amount was reduced to 10 mL compared to arterial administration. Additionally, we observed the patient’s condition for rapid diffusion and safety by avoiding administration over a very short period of time, instead conducting administration over the course of three hours.

The above content was added to the “Materials and Methods” section.

Regarding the INH sample size, this report is a summary of what was conducted since they are the results of an observational study and did not involve patient allocation. We also included in the Discussion that the number of patients needs to be increased, and patients need to be observed in order to make comparisons by administration route.

6. What do 1 and 2 represent in Figure 2? The x-axis naming of all graphs should be corrected. Label in an understandable way when looking at the graph or add a description to the figure legend. It is irrational for the "error bar SE" label to be on the x-axis. Add this to the figure legend and label it in a way that makes your data understandable.

The “1” and “2” in Fig. 2 were specified as “before administration” and “after administration”.

The X-axis was clarified as the CRP value.

An explanation of the error bars is also provided in the legend.

Discussion

7. The findings should be supported by literature data in the discussion. Possible safety and side effects should be discussed with possible reasons, considering variables such as administration routes, CM source used, disease, etc.

The basis for selecting the target disease from the results of the comparative experiment of AD-MSC-CM and UC-MSC-CM is shown together with the literature data.

We judged that the drug could be safely administered with no observed side effects regardless of the administration route, CM source, or target disease, which we described in the Discussion.

We only evaluated safety in this study, so the sample size was insufficient for comparing differences in effectiveness due to different administration routes for each disease. We have also added that future study is necessary for determining whether similar effects could be achieved using methods that place less of a burden on patients.

---

## [Decision Letter · Decision Letter 1]

29 Jan 2025

Dear Dr. Inami,

Thank you for submitting your manuscript to PLOS ONE. After careful consideration, we feel that it has merit but does not fully meet PLOS ONE’s publication criteria as it currently stands. Therefore, we invite you to submit a revised version of the manuscript that addresses the points raised during the review process.

The answers of the Authors to the reviewers are almost all largely insufficient. In particular, n+but not only, the answer: "We judged that the drug could be safely administered with no observed side effects regardless of the administration route, CM source, or target disease, which we described in the Discussion.

We only evaluated safety in this study, so the sample size was insufficient for comparing differences in effectiveness due to different administration routes for each disease. We have also added that future study is necessary for determining whether similar effects could be achieved using methods that place less of a burden on patients" means that this study is extremely limited and does not merit publication.

We look forward to receiving your revised manuscript.

Kind regards,

Gianpaolo Papaccio, M.D., Ph.D.

Academic Editor

PLOS ONE

Additional Editor Comments:

The answers of the Authors to the reviewers are almost all largely insufficient. In particular, n+but not only, the answer: "We judged that the drug could be safely administered with no observed side effects regardless of the administration route, CM source, or target disease, which we described in the Discussion.

We only evaluated safety in this study, so the sample size was insufficient for comparing differences in effectiveness due to different administration routes for each disease. We have also added that future study is necessary for determining whether similar effects could be achieved using methods that place less of a burden on patients" means that this study is extremely limited and does not merit publication.

In addition the pitfalls contained in it as outlined by one reviewer lead to the thought that this is the last possibility they have to effectively amend the paper.

Reviewers' comments:

Reviewer's Responses to Questions

**Comments to the Author**

Reviewer #1: (No Response)

Reviewer #2: All comments have been addressed

2. Is the manuscript technically sound, and do the data support the conclusions?

Reviewer #1: Partly

Reviewer #2: Yes

3. Has the statistical analysis been performed appropriately and rigorously?

Reviewer #1: Yes

Reviewer #2: Yes

4. Have the authors made all data underlying the findings in their manuscript fully available?

Reviewer #1: Yes

Reviewer #2: Yes

5. Is the manuscript presented in an intelligible fashion and written in standard English?

Reviewer #1: No

Reviewer #2: Yes

Reviewer #1: 1. Overall, the new articles have improved the focus on the objective in the introduction. However, they should be presented in a more comparative and cohesive way, not just listed as separate studies. Improving the flow and removing repetitive content would make the section clearer and more concise.

2. The headings presented under the material method are mixed. This should be presented in a logical flow. Ethics should be at the beginning, statistics should be at the end, the dosing schedule should be followed by dosing methods, etc. The titles under the dosing method section should be structured as subheadings. However, another distinct method appears later, disrupting the organization. The section should be properly arranged to ensure clarity and consistency.

3. Although the article has an interesting topic and method, it needs improvement in writing and presentation of data.

Reviewer #2: (No Response)

**Do you want your identity to be public for this peer review?** For information about this choice, including consent withdrawal, please see our Privacy Policy

Reviewer #1: No

Reviewer #2: No

---

## [Author Response · Author response to Decision Letter 2]

20 Mar 2025

Response to Reviewer’s Comments

Reviewer Comments to Author

1. Overall, the new articles have improved the focus on the objective in the introduction. However, they should be presented in a more comparative and cohesive way, not just listed as separate studies. Improving the flow and removing repetitive content would make the section clearer and more concise.

Reply:

Thank you for your valuable feedback. We have revised the introduction to present the referenced studies in a more comparative and cohesive manner, improving the overall flow and removing redundant content. The therapeutic effects of conditioned medium (CM) have now been organized by target disease, highlighting its advantages over stem cell therapy. Additionally, to maintain focus on inflammation, we excluded chronic pain patients who did not undergo blood collection and whose CRP levels could not be assessed. To enhance clarity, we, have also updated the manuscript title to:

"Safety assessment of multiple systemic administration of human mesenchymal stem cell-conditioned medium for various chronic diseases."

2. The headings presented under the material method are mixed. This should be presented in a logical flow. Ethics should be at the beginning, statistics should be at the end, the dosing schedule should be followed by dosing methods, etc. The titles under the dosing method section should be structured as subheadings. However, another distinct method appears later, disrupting the organization. The section should be properly arranged to ensure clarity and consistency.

Reply:

Thank you for your insightful comments. We have reorganized the Methods section for improved clarity and logical flow. Ethics is now positioned at the beginning, followed by the production methods and properties of CM. The administration schedule now precedes the administration methods, with titles properly structured as subheadings. Statistics have been placed at the end, and the previously disjointed section has been integrated for better coherence.

3. Although the article has an interesting topic and method, it needs improvement in writing and presentation of date.

Reply:

Thank you for your feedback. We have revised the manuscript to enhance clarity and ensure effective data presentation. Specifically, we have emphasized that safety was evaluated based on the absence of CRP level increases, indicating no progression of inflammation.

---

## [Editor Report · Decision Letter 2]

23 Mar 2025

Safety assessment of multiple systemic administration of human mesenchymal stem cell-conditioned medium for various chronic diseases

PONE-D-24-23481R2

Dear Dr. Inami,

We’re pleased to inform you that your manuscript has been judged scientifically suitable for publication and will be formally accepted for publication once it meets all outstanding technical requirements.

Kind regards,

Gianpaolo Papaccio, M.D., Ph.D.

Academic Editor

PLOS ONE
---

## [Editor Report · Acceptance letter]

PONE-D-24-23481R2

PLOS ONE

Dear Dr. Inami,

I'm pleased to inform you that your manuscript has been deemed suitable for publication in PLOS ONE. Congratulations! Your manuscript is now being handed over to our production team.

Kind regards,

on behalf of

Prof. Gianpaolo Papaccio

Academic Editor

PLOS ONE